# Global meta-analysis shows reduced quality of food crops under inadequate animal pollination

Elena Gazzea[1] ✉, Péter Batáry[2] & Lorenzo Marini [1]

Animal pollination supports the production of a wide range of food crops fundamental to maintaining diverse and nutritionally balanced diets. Here, we present a global meta-analysis quantifying the contribution of pollination to multiple facets of crop quality, including both organoleptic and nutritional traits. In fruits and vegetables, pollinators strongly improve several commercially important attributes related to appearance and shelf life, whereas they have smaller effects on nutritional value. Pollination does not increase quality in stimulant crops, nuts, and spices. We report weak signals of a pollination deficit for organoleptic traits, which might indicate a potential service decline across agricultural landscapes. However, the deficit is small and non-significant at the α = 0.05 level, suggesting that pollen deposition from wild and/or managed pollinators is sufficient to maximise quality in most cases. As producing commercially suboptimal fruits can have multiple negative economic and environmental consequences, safeguarding pollination services is important to maintain food security.

Animal pollination plays an essential role in flowering plant reproduction[1–3], supporting a wide share of cultivated food crops such as fruits, vegetables, nuts, and spices[4,5]. Although several staple crops are wind- or self-pollinated, many animal-pollinated crops are rich sources of micronutrients contributing to diverse and nutritionally balanced diets[6–8]. In the last decades, multiple anthropogenic pressures have threatened the diversity and abundance of pollinators[9]. This pollinator decline coupled with the global expansion of pollination-dependent crops[10] might indicate a growing risk of pollination deficits worldwide, threatening yields and stability of agricultural production[11], as well as multiple aspects of human health[8].

In the attempt to raise awareness of such alarming trends and to suggest targeted priority actions to reverse the ongoing loss of pollinators, much of the recent research has focused on quantifying pollinator contribution to food crop production[12–14]. Several global analyses across multiple cropping systems have demonstrated the key role of managed and wild pollinators in enhancing crop yield[5,15–19] and its spatial and temporal stability[11,20]. However, the available synthesis studies have focused on the pollination effect on yield-related

measures, such as fruit set or seed weight, while a comprehensive global assessment of the role of pollination in enhancing multiple aspects of food quality is still lacking (but see[21]).

Food quality is a multi-dimensional concept with several attributes influencing a product value[22]. Food quality is often inferred from sensory characteristics (e.g., appearance) and health perception, such as its nutritional profile[23]. These extrinsic and intrinsic quality traits are interlinked with the perception of food safety, affecting quality standards along the supply chain[24] and consumers' purchase behaviour[25]. As food quality can affect market prices and the behaviour of many actors along the food supply chain, it is crucial to quantify the role of animal pollination in determining food quality and marketability. Fortunately, there is a large body of empirical research available for a large number of crops worldwide. The aim of this study is to provide a global quantitative review on the effects of animal pollination on several aspects of food quality including both organoleptic characteristics and nutritional value. We applied a systematic literature review approach to identify relevant studies, followed by a set of multi-level meta-analyses to estimate the contribution of animal pollination

[1]Department of Agronomy, Food, Natural resources, Animals and Environment (DAFNAE), University of Padua, Legnaro (Padua), Italy. [2]"Lendület" Landscape and Conservation Ecology, Institute of Ecology and Botany, Centre for Ecological Research, Vácrátót, Hungary. ✉e-mail: elena.gazzea@unipd.it

to several quality traits of 48 globally important crops. Here, we show that animal pollination greatly improves organoleptic and marketability traits of food crops and, to a lesser extent, their nutritional values. In most cases, current activity of wild and/or managed pollinators is sufficient to ensure optimal food quality. However, we report weak signals of pollination deficits across agricultural landscapes, which encourage the adoption of pollinator conservation actions.

## Results

### General literature patterns

We based our analyses on 1197 effect sizes from 153 publications for pollination service (i.e. difference between open pollination and pollinator exclusion), and 682 effect sizes from 86 publications for pollination deficit (i.e. difference between hand pollination and open pollination) (Fig. 1). We did not find a strong geographical bias in the distribution of the studies (Fig. 2), that were performed in 48 countries. Studies were published between 1968 and 2023, with the number of publications considerably increasing over time (Supplementary Fig. 1). We included studies performing pollination experiments across 48 different crops (Supplementary Table 1). Although our literature review encompassed all animal pollinators, only one study[26] specifically focused on vertebrate pollinators, while most studies tested the effects of single insect species or entire pollinator communities, in which it was not possible to know species identities. In total, selected studies explored the effects of 59 single insect species (Supplementary Table 2). Experiments were performed both under field and greenhouse conditions. Most studies considered organoleptic traits while nutritional value was less investigated (Supplementary Fig. 2; Supplementary Fig. 3).

### Effect of pollination service on quality traits

First, we tested the effect of pollination service comparing quality scores between open-pollinated vs. pollinator-excluded plants. Animal pollination improved the overall quality of crops by 23% (95% confidence intervals, hereafter CI = 16%: 30%; $p < 0.001$) (Fig. 3a; Supplementary Fig. 4). The positive effect of pollination service varied between quality traits (Likelihood Ratio Test, hereafter LRT, $p < 0.001$; $R^2_{marginal} = 0.062$). In particular, pollination contributed in enhancing fruit organoleptic traits up to 27% (CI = 20%: 34%), while fruit nutritional traits were less influenced by animal pollination (mean = 7%, CI = 1%: 14%) (Fig. 3a). A more detailed classification of quality traits revealed that pollinators mostly improved size, shape, and commercial grade, while firmness and micronutrients were improved with lower

statistical support (LRT $p < 0.001$; $R^2_{marginal} = 0.090$) (Supplementary Fig. 5). A model with pollinator group as moderator did not differ from the null model (LRT $p = 0.312$) (Supplementary Table 3), revealing that all pollinator groups affected quality equally (Fig. 3a). Similarly, pollination service benefited both fruits and vegetables equally (LRT $p = 0.344$) (Fig. 3a). We did not find evidence that pollination service benefits varied with the scale at which experiments were conducted (LRT $p = 0.296$), cropping environment (field vs. greenhouse) (LRT $p = 0.335$), nor climatic region (LRT $p = 0.950$).

### Effect of pollination deficit on quality traits

Second, we tested the presence of a pollination deficit by comparing quality scores between open-pollinated vs. hand-pollinated plants, assuming that hand pollination can provide optimal pollen deposition. We did not detect an overall pollination deficit on food quality (mean = 2%, CI = −2%: 5%; $p = 0.390$) (Fig. 3b). However, the effect of hand pollination varied between quality traits (LRT $p < 0.001$; $R^2_{marginal} = 0.031$). We found a weak positive effect of hand pollination on organoleptic traits, improving the size of food crops (test of moderators with the null hypothesis that individual coefficients were zero, hereafter TM, $p = 0.054$), and a negative effect of hand pollination on nutritional traits, particularly on macronutrients (TM $p = 0.045$) and, with lower statistical support, on micronutrients (TM $p = 0.085$) (Supplementary Table 3, Supplementary Fig. 6). The specification of pollinator group as moderator did not improve the model (LRT $p = 0.212$). The positive effect of hand pollination was more evident in fruit crops (TM $p = 0.056$), than in other types of crops (LRT $p = 0.336$) (Fig. 3b). We did not find evidence of a pollination deficit varying with the experimental scale (LRT $p = 0.334$), cropping environment (field vs. greenhouse) (LRT $p = 0.550$), nor climatic regions (LRT $p = 0.215$).

## Discussion

A large body of research has demonstrated that maintaining optimal pollination services is essential to achieve high and stable crop yields[11,15–18,20,27,28]. In our meta-analysis, we quantified the effect of pollination on food quality at the global scale and across all major food crops. Animal pollination service strongly improved multiple organoleptic and commercially important traits of fruits and vegetables, while it contributed to a lesser extent to food nutritional content. Using hand pollination as benchmark, we found a weak sign of pollination deficit for a few traits that might indicate a signal of service decline across agricultural landscapes. However, the observed deficit was small and with weak statistical support, suggesting that in most cases pollen deposition from wild and/or managed pollinators is sufficient to maximise quality.

We showed that approximately one-fourth (23%) of the quality across 48 different crops important in human diet is owed to pollination performed by animals. The observed benefits did not depend on climate, experimental approach, cropping environment, nor pollinator group. However, we observed large variability among quality traits, with animal pollination greatly enhancing commercially important traits, such as appearance and storability, and contributing less strongly to increasing food nutrients. Pollinator-related improvements in marketable and storable quality are likely directly determined by phytohormonal processes activated by fertilisation success[29]. In particular, successful ovule fertilisation triggers an auxin-mediated promotion of gibberellins, which regulate the simultaneous seeds formation and fruit growth[30]. Thus, optimal pollination promotes fruit development in size, thereby increasing its weight, and preventing malformations. Moreover, the same phytohormones increase pulp firmness, thus increasing the post-harvest quality of fruits[21,31]. The observed marginal increase of some micronutrients, particularly some polyphenols, may be linked to hormonal processes[32] or to defensive responses against overexploitation of flowers by insects[33]. The effects

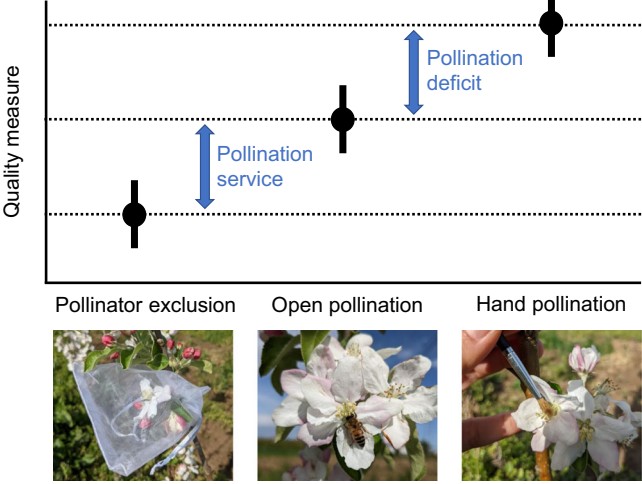

**Fig. 1 | Schematic visualisation of the pollination metrics used to quantify quality change.** Pollination service and pollination deficit are displayed in relation to the commonly performed experimental pollination treatments.

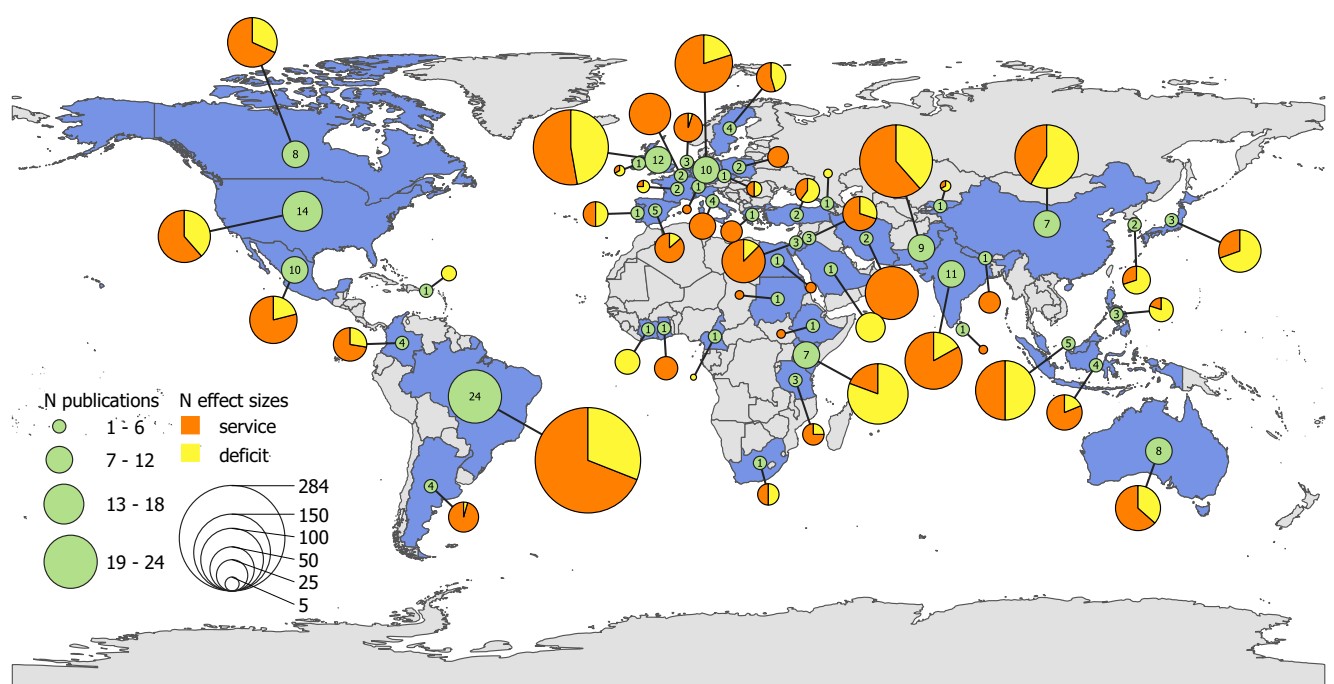

**Fig. 2 | Geographical distribution of the studies.** The map displays in blue the countries for which at least one effect size was included in the meta-analysis. For each included country, the total number of publications and the total number of effects sizes per pollination service (in orange) and pollination deficit (in yellow) is specified, separately.

on organoleptic traits were equally strong in fruits and vegetables, including many high-value crops such as apple, strawberry, pepper, and tomato. On the contrary, the weaker effect observed on micro- and macronutrients is probably related to a dilution effect following the rapid cell expansion and the increase of water content as fully pollinated fruits grow in size[34–36]. The uncertainty linked to the effect of pollination on fruit nutritional values might also depend on the specific synthesis process during fruit formation, and on the high instability of such compounds during fruit metabolic activity[35]. Thus, while fruit size and appearance seem more dependent on pollination than other yield-enhancing factors[37], fruit nutrients are possibly affected also by several environmental factors and/or agronomic practices[38,39].

Beside the effect of pollinators exclusion, we also tested for a potential pollination deficit by comparing quality metrics between animal pollination service and hand pollination. We only detected a weak sign ($p = 0.054$) of a pollination deficit affecting fruit size. Although the effect size was small and with low statistical support, the observed pollination deficits might indicate low levels of landscape-wide activity of pollinators across agricultural landscapes[9]. As emerged from studies focusing on crop yield[18,28,40], the common practice of supplementing pollinator communities by honeybees and other managed pollinators may have contributed to reduce the pollination deficit. In greenhouses and in landscapes dominated by large monocultures of perennial crops, such as apple or almonds, the production of fruits already depends on managed pollinators alone[41–43]. On the contrary, we observed that fruit nutritional qualities were reduced (−4%) when hand pollination was performed. As previously discussed, nutrients can decrease following a pollination level that maximises fruit size through nutrient dilution.

Although we summarised primary data from 190 studies, the experiments tested a large number of crops, pollinator species, and quality traits, and many combinations of moderators were missing or underrepresented. Our literature search highlighted important knowledge gaps in pollination research. First, we could find only a few or no studies exploring the effect of pollination on the quality of spices and stimulant crops with high economic value, such as coffee or cocoa.

Second, we could not test a potential cultivar effect to explain some of the observed within-crop variability. Cultivars are considered an important source of effect size variation[21,34,38], and may have partly caused the observed heterogeneity of our results. However, as farmers adopt different cultivars depending on local conditions and market trends[27], the degree of replication is often insufficient to test this potential important variable. Third, we did not find a comparable effort in studying insects and vertebrates, which are fundamental pollinators especially in tropical regions[44]. Vertebrate pollinators may have been included in our study as part of the pollinator community, but the role of single taxa could not be disentangled. Fourth, potential context-dependency and interactions between pollination and other biotic and abiotic factors are still largely unknown[45,46]. Hence, more manipulative research is needed to explain the variability in the pollination effect on food crop quality.

Our findings have important implications for both agriculture and food industry. Following the rapid expansion and the internationalisation of the food sector, public and private regulations on food quality and safety have been adopted worldwide, most of which are based on appearance and perishability of products[24]. Hence, the production of commercially suboptimal fruits and vegetables substantially affects growers' harvest decision-making, and consequently their access to fresh produce markets or alternative processing[47]. Furthermore, research on purchase behaviour highlights a systematic rejection of unprocessed products that deviate from normality[25,48]. Only a few studies focusing on single crops have attempted to estimate pollinators' commercial value integrating both yield and quality benefits[21,34]. For instance, thanks to their improvements in fruit weight and shelf life, pollinators have been estimated to contribute up to 2.90 billion US$ due to the pollination of commercialised strawberries in the EU in a single year[21]. As quality perception has a strong subjective component[23], calculations should consider not only the value of tangible traits, but also integrate the value of specific traits meeting consumers' beliefs and expectations[49]. Furthermore, poor marketable and storable quality increases food waste along the supply chain[50]. The relationship between pollination and food waste has been almost completely ignored[31], but it

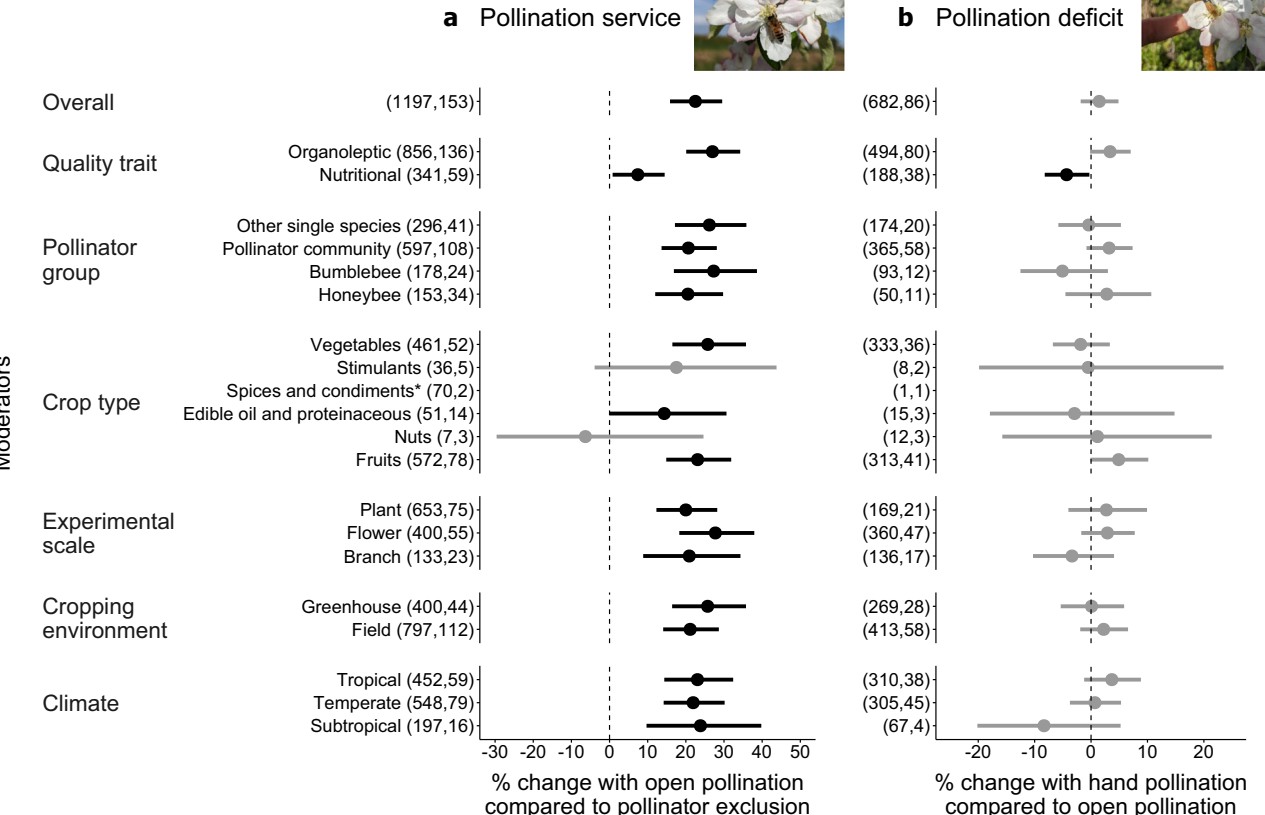

Fig. 3 | **Change in food crop quality for different moderators used in the pollination service (a) (k=1197 effect sizes) and pollination deficit (b) (k = 682 effect sizes) datasets.** Bars around the means indicate 95% CI. Bars not intersecting with zero (dashed line) are coloured in black and indicate a statistically significant change. Estimates are derived from models without the intercept to test within-group differences. The overall pollination outcome was estimated as the pooled effect size (null model). The first and second numbers in parentheses following each moderator level name indicate, respectively, the number of effect sizes used to derive the displayed statistics, and the number of studies included in each calculation. *To improve figure clarity, non-significant effect sizes for spices and condiments are not shown as they exceed figure limits (Mean (95% CI) for pollination service: 27.8% (−11.3%: 84.1%); and pollination deficit: 73.0% (−19.5%: 271.5%). For a full figure displaying all effect sizes for each moderator level refer to Supplementary Fig. 4.

can have important economic and environmental ramifications. Waste of nutritionally-rich food fosters an already suboptimal global consumption of healthy food[51] and can weight on the rate of land conversion to agriculture and/or on the degree of agricultural intensification in view of rapidly increasing global food demand[52]. As the deterioration of animal pollination services cause losses in yield[28], yield-dependent nutritional deficiency[7], and threatens crop marketable quality, there is the urgent need to adopt effective local and landscape management strategies to increase floral and nesting resources and to reduce current environmental pressures on wild and managed pollinators across agricultural landscapes.

## Methods
### Systematic literature survey
Our literature survey aimed at identifying studies experimentally assessing the effect of pollination treatments on the quality of crops important in the human diet. First, we conducted a systematic literature survey in Scopus and ISI Web of Science (WoS) Core Collection (SCI-EXPANDED index) databases. We combined two separate literature searches using two different strings, as during the peer review process we were asked to try a different search approach. Both searches followed the PICO (Population Intervention Comparator Outcome) framework, including terms related to crops, pollinators, and quality. A first literature search was conducted on the 28th of January 2021 and included studies from 1960 (Scopus) and from 1985 (WoS) to 2021. We used the following search string within titles, abstracts, and keywords:

crop* OR *bean* OR palm OR tomato OR banana OR *melon OR *apple OR grape* OR citrus OR cucumber OR brassica OR *nut OR mango OR eggplant OR sunflower OR safflower OR plantain OR pepper OR pumpkin OR squash OR broccoli OR peach OR pear OR *peas OR olive OR papaya OR plum OR coffee OR okra OR asparagus OR date OR *berry OR avocado OR lentil* OR sesame OR cacao OR persimmon OR *fruit OR apricot OR almond OR linseed OR cherry OR artichoke OR pistachio OR lupin* OR fennel OR fig* OR mustard OR quince* OR *currant OR hops OR poppy
AND
*pollinat* OR *bee OR *bees OR hover*
AND
qualit*.
A second literature search was conducted on the 23rd of February 2023 and included studies from 1960 (Scopus) and from 1985 (WoS) to 2023. We used the following search string within titles, abstracts, and keywords:
crop OR fruit
AND
pollinat* OR *bee OR *bees OR hover* OR bird* OR bats OR bat OR avian OR chiroptera* OR lorikeet* OR flowerpecker* OR

honeyeater* OR whiteeye* OR warbler* OR hummingbird* OR sunbird* OR nectariv* OR "nectar feeding" OR "flying fox*" OR lemur* OR possum* OR lizard* OR squamata OR iguania OR gekkota OR gecko* OR rodent* OR gerbil OR mammal*
AND
qualit*.

Second, to include also grey literature, we conducted two parallel searches in Google Scholar (location of the search: Padova, Italy). We used the following query within all fields in both 2021 and 2023:

pollination quality crop OR fruit.

For both Google Scholar searches, we sorted results by relevance and checked the first 600 publications[53]. Our literature survey included peer-reviewed articles, conference proceedings, book chapters, and published theses. To minimise language biases, relevant non-English publications with an abstract in the English language were included throughout the literature search[54].

To ensure reproducibility, we reported detailed information regarding both 2021 and 2023 literature searches as a PRISMA (Preferred Reporting Items for Systematic Reviews and Meta-Analyses) statement[55] in Supplementary Figs. 7 and 8. Additional details on all the meta-analysis steps are reported in a PRISMA-EcoEvo checklist[56] (Supplementary Data 1).

### Definition of pollination metrics

We quantified the pollination effect using two metrics:[57] (1) pollination service, i.e. the difference between animal pollination and animal pollination exclusion; (2) pollination deficit, i.e. the difference between animal pollination and the maximum potential pollination provided by supplementing pollen with hand pollination (Fig. 1). For crops that benefit from outcrossing, the level of fecundity achieved by hand pollination with outcross pollen should represent perfect pollination. By manually providing large quantities of pollen, saturation should occur (i.e. all ovules are fertilised), while this might not always be the case with animal pollination[58]. To calculate these metrics, the pollination treatments performed in the studies included in the meta-analysis (see next section for details about the selection process) were classified into three categories: (1) open pollination, when pollinating animals were free to visit flowers; (2) hand pollination, when cross pollen was manually applied to individual flowers or supplemented to the pollen carried by animals; (3) exclusion treatment when pollinating animals were completely excluded from visiting flowers but wind pollination was allowed.

### Literature selection process and effect size calculation

We performed an initial title and abstract screening in which publications having a clearly different topic were excluded. In the selection process, we included all animals that were tested as potential pollinators. Potentially relevant records with accessible full-text were examined to assess whether the following inclusion criteria were met: (1) studies reported results derived from primary data of manipulative experiments. Hence, meta-analyses and qualitative syntheses, such as reviews and books were excluded at this stage; (2) studies measured at least one quality trait of the edible parts of the studied crops (see Supplementary Table 4 for a list of quality traits included). We excluded publications reporting solely fruit set, yield, seed germination, seed set, or seed number; (3) experiments included at least two different pollination treatments, so one of the two pollination metrics of interest (either pollination service or deficit) could be calculated; (4) there was fruit production under all relevant treatments, as quality can be assessed from developed fruits only. Eligible studies were supplemented with their relevant references, and with references from published quantitative and qualitative synthesis studies retrieved from the systematic literature search and initially excluded from our dataset by criterion (1).

From each selected publication, we extracted the mean, standard deviation, and number of replicates for each quality trait. Values were either extracted from text, supporting information, tables, figures[59,60], or calculated from raw data, when available. If not explicit, standard deviation was derived from standard error, confidence intervals, or range[61]. Missing data were requested directly from the authors of papers dated 2000 onwards (97 publications). Whenever possible, the negative quality measures were converted to their corresponding positive ones (e.g., the percentage of malformed fruits was converted to the percentage of regular fruits). Data was structured and coded using a Microsoft Excel spreadsheet.

As a measure of effect size, we calculated the magnitude of the animal or hand pollination effect on crop quality using the natural logarithm of the response ratio (lnRR)[62] following Eq. (1):

$$lnRR = \left( \ln \frac{X_{treatment}}{X_{control}} \right) \qquad (1)$$

where $X_{treatment}$ and $X_{control}$ are the means of the quality traits measured respectively in the open pollination and pollinator exclusion treatments for pollination service, and in the hand pollination and open pollination treatments for pollination deficit. A positive lnRR value indicates an increase in quality with animal or hand pollination. To help visualise the results, the mean percentage of quality change was calculated following Eq. (2):

$$\% \text{ change} = \left( e^{lnRR} - 1 \right) * 100 \qquad (2)$$

In this study, we allowed only for weighted meta-analyses, thus we excluded effect sizes with missing variance. In the meta-analysis models described below, we weighted the effect sizes by their inverse variance. The integration of both our literature searches resulted in a total number of 190 studies, which were used in the following analyses (Supplementary Fig. 8).

### Moderators

For each study, we extracted the following moderators: (1) quality trait; (2) pollinator group; (3) crop type; (4) experimental scale; (5) cropping environment; (6) climate; and (7) year of publication. We assessed the potential association among categorical moderators using mosaic plots from the vcd package[63]. The analysis allows the flat representation of a matrix of pairwise contingency tables, depicted through mosaic and association plots and help to visualise potential collinearity among moderators (Supplementary Fig. 2; Supplementary Fig. 3).

**Quality trait.** Due to its intrinsic subjective component related to consumers' perception, food quality is not uniquely defined[64]. In this study, we considered as part of food quality those attributes describing physical properties (e.g., diameter, weight, pulp firmness), nutritional value (e.g., sugars, vitamins, minerals), stability (shelf life), marketability (e.g., commercial grade), and user-oriented quality traits, such as appearance and taste. We pooled quality traits into two broad categories: (1) organoleptic and (2) nutritional characteristics. A more detailed classification was further developed into seven quality categories: (1) size; (2) shape; (3) external appearance and taste; (4) firmness; (5) commercial grade; (6) macronutrients; (7) micronutrients (Supplementary Table 4).

**Pollinator group.** We included studies testing the effect of both single pollinator species and whole pollinator communities. To test the effect of pollinator identity, we grouped the studies into four broad categories: (1) honeybee, with insects belonging to the *Apis* genus; (2) bumblebee, with insects belonging to the *Bombus* genus; (3) pollinator community, when there was an open pollination treatment and the exact pollinator species identity could not be inferred; (4) other single

species, including species that were individually tested, and for which it was not possible to create separate categories due to small sample size (Supplementary Table 2).

**Crop type.** Based on the categories adopted by Klein et al[5]., we categorised the edible crops included in our analyses into six broad groups: (1) vegetable crops; (2) fruit crops; (3) nut crops; (4) edible oil and proteinaceous crops; (5) stimulant crops; (6) spices and condiments. As the large majority of effect sizes were representative of vegetable and fruit crops, we present mosaic plots with the other moderators only for those (Supplementary Table 1).

**Experimental scale.** Experimental manipulation level is considered as an important variable when assessing the effect of pollination on crop yield, quality, and stability[20,65]. For each study, we defined the manipulation level at which the pollination experiment was conducted: (1) flower, when bagging treatment or hand pollination was applied at the flower level; (2) branch, when the experiment was performed on only part of the plant; (3) plant, when pollination treatments were applied to the whole plant.

**Cropping environment.** This variable refers to the experimental plant growing environment: (1) field, when plants were grown in open field under natural conditions; (2) greenhouse, when plants were grown under a controlled environment.

**Climate.** Each country was categorised in one of three climatic zones based on latitude: (1) tropical (0°–23°); (2) subtropical (23°–33°); (3) temperate (>33°). We used latitude as a proxy for different macro-climatic conditions.

**Year of publication.** To test for a time-lag bias, i.e. the tendency of finding larger effect sizes in earlier published studies than those of later studies[66], we included publication year as a moderator. However, also the contrary is possible where a temporal increase in effect sizes could be related to a true decline in pollination service related to increasing environmental pressures on pollinators[3].

## Multi-level meta-analysis models
Selected publications often performed more than one trial, for instance over different years, using different pollinator species, or measured multiple quality traits, and thus reported more than one effect size. This clustering of effect sizes at any organizational scale violates model assumptions of independence and can affect the overall meta-analytic estimates[67]. To examine the variation in effect sizes, and to account for non-independence of observations, we used multi-level meta-analytical models, which are equivalent to linear mixed-effects models[68,69]. We accounted for clustered effect sizes by including random (nesting) factors. Additionally, by specifying variance-covariance matrices between effect sizes and sampling errors, we accounted for their potential non-independence originated when multiple outcomes were measured from the same experimental unit. For constructing variance-covariance matrices we assumed a sampling correlation of 0.5 among clustered effect sizes. Finally, to account for correlated effect sizes' sampling errors when different trials shared a treatment (e.g., a different species of pollinator compared to the same pollinator-exclusion group), we used a modified number of replicates in the effect sizes' variance calculation, by dividing the sample size of the control group by the number of times the control was shared[70,71].

To estimate the overall effect of pollination on quality traits, we first constructed null models containing only random effects. We compared null models with the following terms combined either as crossed or nested random terms: (1) publication unique identifier; (2) country where the study was conducted; (3) year of the experiment. Additionally, in all of the null models tested, we included a unique identifier per effect size/data row as random term, to estimate the residual heterogeneity[70]. We evaluated the goodness of fit of candidate random structures in null models using Akaike information criterion (AIC) (Supplementary Table 5). The identified optimal random structure contained the unique publication identifier and the year of experiment as crossed terms in the pollination service model, and the unique publication identifier in the pollination deficit model.

To explain heterogeneity in effect sizes, we incorporated moderators as fixed effects in the models with the optimal random effect structure identified in the previous step. We tested the following moderators: (1) quality trait, (2) pollinator group, (3) crop type, (4) experimental scale, (5) cropping environment, and (6) climate. The two-way contingency tables revealed collinearity among pollinator groups, crop types, and cropping environment (Supplementary Fig. 2; Supplementary Fig. 3). In particular, bumblebees and other single pollinator species were mostly tested in greenhouses. Instead, the natural pollinator community was more frequently tested in fruit orchards under field conditions. Also, in pollination service experiments, nutritional quality was investigated mainly in fruit crops. Due to several incomplete combinations, we ran separate models for each moderator variable, and did not test for interactions. We used AIC and likelihood ratio test (LRT) to compare models including moderators with the null model. We reported the results of the omnibus test and interpreted the model coefficients and confidence intervals of each moderator level, separately. To show how much variability was explained by significant moderators, we calculated marginal $R^2$[72]. Then, we examined the significance of variation in effect sizes attributed to each moderator variable using Q statistics[73]. We used maximum likelihood ML to compare models, and restricted maximum likelihood REML to estimate mean effect sizes and their variances in the best models. To visualise model results, we displayed the overall mean effect alongside with 95% confidence intervals (CI). For all analyses, we used the rma.mv function of the metafor package[74] of R software[75] (Supplementary Note).

## Publication bias and sensitivity analysis
To test for publication biases in our dataset, we used different approaches. First, we visually inspected funnel plots of residuals against their precision[68] and did not detect extreme patterns (Supplementary Fig. 9). Second, we used a modified Egger's regression method with an effective sample size to test for publication bias in the log response ratio[76]. This method can provide a test for funnel plot asymmetry and can handle dependence in the effect sizes. The test found no bias in both pollination service and deficit datasets (Supplementary Table 6). Third, to identify potential outliers in our dataset, we calculated hat values and standardised residuals. Effect sizes greater than two times the average hat value and standardised residual values exceeding 3.0 are considered influential outliers[77]. Hat values and residuals inspection did not show any outlier in our datasets (Supplementary Fig. 10). Fourth, by including publication year as a moderator in the models, we tested for time lag bias[66]. We did not detect a time lag bias neither in pollination service effect sizes (LRT $p = 0.415$) (Supplementary Fig. 11a), nor in the deficit effect sizes (LRT $p = 0.437$) (Supplementary Fig. 11b).

To evaluate the robustness of our results, we repeated all the analyses removing the studies contributing more than 5% in each dataset (1 study with 69 effect sizes for pollination service, 2 studies with 92 effect sizes for pollination deficit). Our sensitivity analysis revealed that removing these studies did not change the results (Supplementary Table 7). Additionally, we diagnosed influential observations using Cook's distance[78], and excluded data points when Cook's distance was more than 4/n, where n is the number of observations for estimated outcomes[79]. The effects of identified strong influential points (14 effect sizes for pollination service, 12 effect sizes for pollination deficit) (Supplementary Fig. 12) on the pooled effect size were rather minor with similar magnitude and direction of effect

sizes and their 95% CI (Supplementary Table 7). Moreover, we examined the extent to which the pollination service and deficit effects were sensitive to the assumption of correlated sampling errors and effect sizes[80]. Assuming a 0.8 correlation did not significantly change the overall estimates (Supplementary Table 7). Lastly, we performed a sensitivity analysis by clustering the effect sizes at the level of individual studies for the construction of variance-covariance matrices and we did not detect significant changes in the overall estimates (Supplementary Table 7).

### Reporting summary
Further information on research design is available in the Nature Portfolio Reporting Summary linked to this article.

## Data availability
The data used in this study have been deposited on Zenodo digital repository under https://doi.org/10.5281/zenodo.8113788. Literature search databases used in this study are: ISI Web of Science (WoS) Core Collection; Scopus; Google Scholar.

## Code availability
The code used in this study has been deposited on Zenodo digital repository under https://doi.org/10.5281/zenodo.8113788.

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

## Acknowledgements

The study was supported by the EU Horizon 2020 329 project "Safeguard" (Grant agreement ID: 101003476) funded under Societal Challenges - Climate 330 action, Environment, Resource Efficiency and Raw Materials and by the OLEO.BEE project (cod. 2105-0003-1463-2019) funded by the Veneto Region to LM. PB was supported by the Hungarian National Research and Development and Innovation Office (NKFIH KKP 133839).

## Author contributions

E.G. and L.M. conceived the study; E.G. performed the literature search and extracted effect sizes; E.G. and L.M. analysed the data with inputs from P.B.; E.G. and L.M. wrote the manuscript; All authors discussed the results and contributed to the final draft.

## Competing interests

The authors declare no competing interests.
