## [Peer Review File · Nature Communications]

nature portfolio

Peer Review FileReviewer comments, first round

Reviewer #1 (Remarks to the Author):

This paper presents a systematic review and a set of multi-level meta-analyses to estimate the contribution of animal pollination to several crop quality traits worldwide. The topic and focus are relevant to the field and novel. A quantitative synthesis on pollinator contribution to quality traits is not yet available and could be highly impactful given their impact on commercial value, trade and the relevance to farmers income and livelihoods.

However, I have major concerns on parts of the methodology that I believe need addressing before this research can be published. The statistical analysis is robust, but there are many aspects of the literature search strategy that in my opinion may have prevented from exhaustively capturing relevant studies and caused bias. Many aspect, decisions, inclusion/exclusion criteria should be clarified more explicitly to enable transparency and replicability. Furthermore, I believe the discussion should address the key gaps identified in the review (e.g., lack of vertebrate pollinators study) and show how the results of this analysis can inform future research directions. I gave further details below and I hope my contribution will be useful in improving the manuscript.

Fabrizia Ratto

Major comments:

Title

I'd recommend that given the global scope of the study, this is included in the title. Also, I'd recommend stating in the title that this is a quantitative synthesis

Abstract

I would specify in the abstract what quality measure were used. Even just a couple as examples.

Methodology

L180 The authors state that animal-pollinated crops were identified and included depending on their quantity (tonnes) produced worldwide in 2018 and a list is provided in the supplementary material but how were they then selected? Were only the major crops included or all? I see there are some criteria for exclusion of some crops, though explicitly mentioning some crops and not others in the search string may have biased the search. (also, one stated exclusion criterion is that broccoli is excluded as the edible part is an inflorescence, yet it features in the search string..). How about durian, pitaya and other highly pollinator dependent fruits? Perhaps it would have been more effective not to include crop type or use a NOT for those crops which were to be excluded.

For example, Tremlett et al 2020, which seems an article that should be included in the search, is missing. Ratto et al 2021 also seems a suitable study, which was not included. These may have been screened and excluded for valid reasons, I mention these because I know of these studies, but it made me question whether more studies may have been missed.

Linked to the above, another concern regarding bias of the search string is pollinator taxon. The authors do not include what was defined as a pollinator in the study but state that they included insects and other pollinating animals. However, the search string is focused on insects, using pollinat* as the only term to capture anything that isn't a bee or a hoverfly but because the pollinator term is linked with an 'AND' to a selected number of crop types (none of which are vertebrate pollinated), the search would not return any study on vertebrate pollinated crops. Again, Tremlett et al 2020, and Classen et al 2014 are examples of studies that seem suitable for inclusion, but they are not included.

Also, the authors state that "To include potential studies focusing on other crops, general "crop" and "fruit" terms were also added in the search string". However, crop* is only in the first of the three blocks of search, and fruit* only in the last chunk. I would be interested to understand why the search string has been structured in three parts instead of listing all the crop types in one go? i.e. TS= qualit* AND (*pollinat* OR *bee OR *bees OR hover*) AND all the crop types..

Although the authors state that they included non-English literature (not clear whether for the grey

literature only or overall, please clarify) the search was conducted in English, inevitably excluding potential publication in other languages. I appreciate that it may not always be possible to screen in other languages, but for a global analysis this may be necessary.

If there is a sound rationale for all the above points on the search string, it needs to be better clarified in the methods. Otherwise, I'd recommend revisiting the search string and re-run it. I accept that no search string will capture everything, but I believe that as it is, this search strategy has missed relevant articles.

L247 It is not clear how the categories for the moderator "Pollinator group" were chosen. What is the definition of "natural/ambient pollinators"? Does the "natural" category include studies where pollinators were not identified to species/genus level? (this is only mentioned in the results L67 needs to be in methods). If so, I am not sure it is worth having a "other" category. Some of the species in the "other" group may well overlap with natural/ambient species and they would be natural/ambient pollinators too. The only difference between "other" and "natural" seems to be that the latter were not identified, so perhaps it would make more sense to pull them together? were they not even identified by order/genus? Given that the natural community show significant effect on quality in the pollination deficit, it would be interesting to disentangle that. It may be that I am missing something and there is a sound reason for this classification, but the categories need to be defined and the rationale for choosing them be more explicit.

Results

L70-101 - Figure 3 One of the pros of using the log RR is that it can be converted to a percentage, which is more intuitive to the reader. I'd suggest reporting those percentages, at least the key ones in the results. The AIC in itself is not highly informative. I'd also suggest to report the results as percentage in Figure 3, which again would aid interpretation and impact of your key results.

Discussion

What I believe is missing in the discussion are key research gaps. For example, crops that are understudied or not studied at all? were there understudied regions? If really there is nothing on vertebrate pollinators effect on quality, this is an important gap given the economic importance of some of vertebrate pollinated fruits. These should be highlighted along recommendations on future research directions.

Minor comments:

L18 Overall?

L43 (But see)?

L56 the contribution of animal pollination OR the contribution of pollinators (here and throughout the manuscript, e.g. L105-108-114)

L60 wind/self pollination

L131 It would be interesting to expand and explore potential reasons why that is. Is there a threshold of pollination above which supplementary pollination does not affect quality? A saturation point? Is there any literature on this?

L134 to my understanding this is what was defined as 'crop dependence on pollination' and was not considered in this study. How was the 32% calculated?

L172 friendly land uses I a bit general, perhaps this can be expanded. Insect friendly farm management? Landscape scape conservation?

L179 could you add a line to briefly describe the aim of the search and some key words? Something along the line of "we conducted a systematic review....to investigate.." "we used a combination of terms..". To give a clear indication of what the aim of the review is

L180 I'd state the date as from-to. The exact date of last search can be moved to the SI

L182 Is this OR between quality crop and fruit? Clarify

L189 It is not clear to me what an unduplicated relevant reference is, could you clarify this? Did you snowball the reference from relevant papers?

L192 The inclusion criteria should provide a definition of animal pollinators, did you include all? (see major comment above)

L193 Animal pollination dependence OR pollinator dependence

L205 I would move this above to introduce the treatment types before the inclusion/exclusion criteria

L216 spell out the acronym on first use

L222/225 animal pollination or pollinator effect (here and throughout the manuscript)
L262 this is indeed an important moderator, but it would call it "experimental scale" or "experimental manipulation level" rather than spatial scale, which to many would evoke landscape scale vs farm scale etc
L277 replace 'initially' with 'earlier'

Reviewer #2 (Remarks to the Author):

Key results

The manuscript deals with the effect of animal pollination on fruit quality using a meta-analytical approach spanning across various pollination dependent food crops globally. The authors found that pollination generally increases food quality but when broken down into specific characteristics only some traits were influenced positively. All pollinators contributed positively to quality with bumblebees being most effective. The authors also found that crop production generally suffers from a pollination deficit for wild bee communities.

Validity

In general, the authors provide a valid and robust interpretation of the data. However, I have some (major and minor concerns) about the following specific interpretations.

The title appears to be slightly misleading. It might be understood so that nutritional food is only produced by inadequate pollination. Hence, I would recommend a minor revision of the title so that this becomes clearer.

The conclusions of the Abstract (lines 23-26) do only fit parts of the results and the manuscript such as it deals with pollination effects and not with imperfection of fruits leading to food waste.

Line 45: It is not only the consumer but aspects such as shelf life (related to firmness) influences also the value for farmers and retailers.

Line 49: Food quality is not only an ultimate factor for farmers income but also a first-hand factor, for instance such as farmers are paid higher for products of class 1 compared with class 2.

Lines 90-91: This result is somehow trivial such as greenhouse growers supply sufficient managed bees to not run into pollination deficits.

Lines 104-106: This sentence might have to be toned down such as there are a lot of studies that focused on crop quality but not on the scale of a meta-analysis.

Lines 144-146: The discussion would benefit by adding one or two more sentences on this, also including interspecific complementarity between pollinator, such as for instance shown by Brittain et al. 2013, *Global Change Biology* (<https://doi.org/10.1111/gcb.12043>).

150-173: It is not just imperfection but also shorter shelf life (and other potential trait-offs), such as shown by Klatt et al. 2013, *Agricultural and Food Security* and Klatt et al. 2014, *Proceedings B* (both already cited in the reference list). It is not clear if this problem applies for all fruits but as stated in the manuscript earlier, firmness is an essential factor being determined by pollination and firmness is also a main factor for the decay of many agricultural products. Hence, this part of the discussion is completely unilateral towards consumers' behaviour but will have to include also other aspects that come with the reduced quality characteristics shown in the results. This also applies for the concluding sentences in the Abstract.

Significance

These results provide a significant insight for the field of pollination effects on food quality and thereby make an important contribution to our understanding about the importance of pollination for agricultural production.

Data and methodology

The authors provide an impressive dataset derived from very many studies and covering large parts of global crop production areas. Literature research, data inclusion/exclusion are well-motivated and justified. The presentation of the data is of high quality including easy-accessible figures.

Analytical approach

The statistics (from the linear mixed model point of view; please see also 'Your Expertise' section) are valid and well-motivated. I have a few questions concerning some of the analyses.

Line 72: How natural is the pollinator community of 'natural pollinated plants'? Nowadays honeybees are ubiquitous in agricultural landscapes (as it is also stated in lines 140-144). Is there any information available about the proportion of honeybees within the pollinator communities? Maybe this could be added into the models? Also, what was the proportion of honeybees in the natural communities that had decreased pollination services?

Lines 82-83: Looking at figure 3, the difference between bumblebees and the natural community seems hardly be significant (confidence intervals overlapping quite distinctly). It is stated that the difference was tested but no test results are shown.

Lines 125-128: Do you control for a potential bias provided by that specific crops that are pollinated by specific pollinator groups (naturally or artificially provided) have been the focus in the literature more frequently?

Lines 207-209: How were differences in pollination dependence between crop varieties handled? Were differences between varieties included into the calculations (lines 212-214)?

Lines 226-227: Does this mean that specific studies (how many?) and/or specific parameters (which?) had to be excluded?

Lines 272-274: Such as habitats can differ largely also between latitudes, landscape type might be a good additional category, maybe describing the main landscape element in the surrounding of the study (e.g. forest, agriculture, urban, etc.).

Lines 276-278: Based on declining pollinator populations, could this effect not also be vice versa, i.e. larger effects in newer studies, at least for outdoor studies with natural pollinator communities?

Modelling in general: Where models tested for spatial and temporal autocorrelation and singularity?

Lines 330-331: A p-value <0.001 is indicating more than just 'potential bias'.

Suggested improvements

The authors argue with the commercial aspects of crop quality at various places and therefore it would add to the argumentation if monetary values could be provided as well.

Lines 116-119: Because this manuscript will be interesting for a broad readership, the discussion would benefit from some more details here so that the underlying processes can also be understood by non-expert readers on this topic.

Further, it would be good to test for deficits between different groups of crops, for instance seed and oil crops versus fruits versus vegetables. It would also add an interesting aspect to the results

to show pollination deficits in dependence on region, country or even the intensity of agriculture. It is indicated that these results are existing in lines 122-124 (and are partly available in the Supplementary Information). This information would also be interesting in the light of the statement in lines 137-139.

Clarity and context

The manuscript concisely and clearly written. I have some questions on parts of the text that were not clear to me.

Line 33: Maybe change 'life support' to something like 'food provision' or similar.

Line 35 Indicate what kind of deficits.

Line 71-72: In the paragraph before you stated that effects from differences between wind and natural pollination as well as between artificial and natural pollination was assessed. Please clarify the term 'non-pollinated vs. naturally pollinated plants' within this framework.

Line 109: Change 'erosion' against a less strong word, 8 % is more a decline.

Line 113: Should it be not 'quality' itself and not 'quality improvement', because the assumption is that sufficient pollination should be available? Hence 24% of crop quality is provided by pollination or pollination contributes to 24% of crop quality.

Lines 130-131: Does maximised artificial pollination include greenhouse and crops where honeybees or bumblebees have been placed at the field? Was this information available for all studies under open field conditions?

Line 131: You mean exclusion of animal pollinators? Please clarify at this and other places in the manuscript.

Lines 134-136: This would also fit into the results section. Please also provide more information about how this value was calculated.

Lines 139-140: In the beginning of this paragraph and the results, it is stated that there is pollination deficit of 8% based on quality traits and 'natural' pollination but here it is stated that there is about one fourth loss. Or is this meant potentially at a complete absence of pollination?

References

Besides my concerns stated in the different sections above, the cited literature is appropriate.

Your expertise

Meta-analytical statistical approaches are outside my expertise. But I reviewed the statistics from the point of linear mixed modelling.

DETAILED RESPONSE TO COMMENTS

RESPONSE TO THE REVIEWER 1'S COMMENTS

	Reviewer 1's comments	Authors' response
1	- This paper presents a systematic review and a set of multi-level meta-analyses to estimate the contribution of animal pollination to several crop quality traits worldwide. The topic and focus are relevant to the field and novel. A quantitative synthesis on pollinator contribution to quality traits is not yet available and could be highly impactful given their impact on commercial value, trade and the relevance to farmers income and livelihoods.	- Thanks for the appreciation of our work.
2	- However, I have major concerns on parts of the methodology that I believe need addressing before this research can be published. The statistical analysis is robust, but there are many aspects of the literature search strategy that in my opinion may have prevented from exhaustively capturing relevant studies and caused bias. Many aspect, decisions, inclusion/exclusion criteria should be clarified more explicitly to enable transparency and replicability.	- We are grateful to the Reviewer for the useful comments. We have performed a new literature search following the suggestions to capture an even larger number of studies. As explained below, we are sorry that the odd syntax of our original search string might have generated some confusion. We have also provided more details about the critical steps during the study selection in the Methods section (please refer to LL 236-250).
3	- Furthermore, I believe the discussion should address the key gaps identified in the review (e.g., lack of vertebrate pollinators study) and show how the results of this analysis can inform future research directions.	- This is a very good suggestion! We added a new paragraph on study limitations, identified knowledge gaps, and future research directions (please refer to LL 157-173).
4	- I gave further details below and I hope my contribution will be useful in improving the manuscript.	- We thank the Reviewer again for the very insightful and constructive comments.
	Major comments	
5	-Title I'd recommend that given the global scope of the study, this is included in the title. Also, I'd recommend stating in the title that this is a quantitative synthesis	- We have changed the title integrating both Reviewers' suggestions. The new title is: "Inadequate pollination compromises the quality of food crops: A global meta-analysis". This reflects the now significant effect of pollination service also on nutritional value. Although the effect size is much smaller than that of organoleptic traits, we think that the title reflects better the main findings.
6	-Abstract I would specify in the abstract what quality measure were used. Even just a couple as examples	- We have now specified that both fruit organoleptic traits and nutritional value were included (please refer to L 17).
7	-Methodology L180 The authors state that animal-pollinated crops were identified and included depending on their quantity	- By using the general term "Crop" in both the original and the revised string we aimed to capture all crops. The broccoli left in the string was unnecessary, but it did not

	(tonnes) produced worldwide in 2018 and a list is provided in the supplementary material but how were they then selected? Were only the major crops included or all? I see there are some criteria for exclusion of some crops, though explicitly mentioning some crops and not others in the search string may have biased the search. (also, one stated exclusion criterion is that broccoli is excluded as the edible part is an inflorescence, yet it features in the search string..). How about durian, pitaya and other highly pollinator dependent fruits? Perhaps it would have been more effective not to include crop type or use a NOT for those crops which were to be excluded.	affect the search, as it was included with an OR. The same for the other specific crops included in the original string. Following Reviewer’s suggestion, we revised the search string using three blocks separated by AND. The search string structure follows the PICO (Population Intervention Comparator Outcome) framework. We now have a first part including two general terms crop and fruit, a second including the taxa (both vertebrates and invertebrates), and a third including the general term qualit*. NEW STRING: crop OR fruit AND pollinat* OR *bee OR *bees OR hover* OR bird* OR bats OR bat OR avian OR chiroptera* OR lorikeet* OR flowerpecker* OR honeyeater* OR whiteeye* OR warbler* OR hummingbird* OR sunbird* OR nectariv* OR “nectar feeding” OR “flying fox*” OR lemur* OR possum* OR lizard* OR squamata OR iguania OR gekkota OR gecko* OR rodent* OR gerbil OR mammal* AND qualit* We followed the Reviewer’s suggestion to avoid including specific crop types. We performed the search within titles, abstract and keywords. We did not perform the search to all fields as the number of studies using the same string went from ~4’500 to ~160’000 and became too generic. On the one hand, the new search helped to include 25 more studies within the original time span of the study. However, it also missed many studies captured with our original string (see response to Comment 12). On the other hand, the inclusion of 2021, 2022, and 2023 allowed to include many recently published studies on the topic. We re-run the analysis with the new datasets and the results have only slightly changed. It was a very useful exercise to compare the results from two separate search strings and realise that probably the best approach in meta-analyses would be to combine multiple search strings instead of trying to find the “best one”. Thanks! The only change we have made in the new analyses is related to the random structure of the pollination service models. The best random structure for the service null model that included the new data resulted in the term “experimental year” added as a crossed term, rather than nested within the publication. This change in the random structure of the model did not affect the overall effect of pollination service (mean (model with nested term)=22.1%, CI=17.4% : 27.0%); mean (model with crossed term)=22.5%, CI=15.9% : 29.5%)
8	- For example, Tremlett et al 2020, which seems an article that should be included in the search, is missing. Ratto et al 2021 also seems a suitable study, which was not included. These may have been screened and excluded for valid reasons,	- Tremlett et al. 2020 was retrieved but excluded because exclusion treatment did not yield fruits, so fruit quality under this treatment could not be measured. Tremlett et al. 2020 thus only presents results comparing the effect of different animal pollinators, but the metric of our interest (pollination service) could not be computed. Ratto et al.

	I mention these because I know of these studies, but it made me question whether more studies may have been missed.	2021 was published after the date of our original search (January 2021). However, it was retrieved with the updated search string and included in the new analyses.
9	- Linked to the above, another concern regarding bias of the search string is pollinator taxon. The authors do not include what was defined as a pollinator in the study but state that they included insects and other pollinating animals. However, the search string is focused on insects, using pollinat* as the only term to capture anything that isn't a bee or a hoverfly but because the pollinator term is linked with an 'AND' to a selected number of crop types (none of which are vertebrate pollinated), the search would not return any study on vertebrate pollinated crops. - Again, Tremlett et al 2020, and Classen et al 2014 are examples of studies that seem suitable for inclusion, but they are not included.	- We now follow the approach of Ratto et al. 2018 (Front Ecol Environ 2018; 16 (2): 82– 90, doi: 10.1002/fee.1763) by explicitly including the taxa on the search string separated by OR. We used the general terms for vertebrates used by Ratto et al. 2018 but we did not use the families as our search was performed on title, abstract and keywords where Latin names of families are rarely used. In the original string, the general term 'crop' was always included in the search besides all the specific crops and coupled with an AND with pollinat* (please see comment below on the structure of the original string). Hence, the presence of non-vertebrate pollinated crops in the string did not exclude studies on crop pollinated by vertebrates. Anyway, the new string now includes also more terms to maximize the ability to find vertebrate pollinators. - Please see comment above explaining the reason for which we excluded Tremlett et al. 2020 from the dataset. Classen et al. 2014 was originally included, but some treatments were initially misunderstood and mistakenly excluded from the dataset. We have now included this study and all the treatments.
10	- Also, the authors state that “To include potential studies focusing on other crops, general “crop” and “fruit” terms were also added in the search string”. However, crop* is only in the first of the three blocks of search, and fruit* only in the last chunk. I would be interested to understand why the search string has been structured in three parts instead of listing all the crop types in one go? i.e. TS= qualit* AND (*pollinat* OR *bee OR *bees OR hover*) AND all the crop types..	- Our original string was divided in three chunks to fit both Scopus and Web of Science website search engine. This is because Scopus does not allow to include more than 256 characters per search field on their website. Hence, the first part is repeated in the three fields (marked in blue). Original string (three fields) FIRST FIELD: qualit* AND (*pollinat* OR *bee OR *bees OR hover*) AND (crop* OR *bean* OR palm OR tomato OR banana OR *melon OR *apple OR grape* OR citrus OR cucumber OR brassica OR *nut OR mango OR eggplant OR sunflower OR safflower OR plantain OR pepper) OR SECOND FIELD: qualit* AND (*pollinat* OR *bee OR *bees OR hover*) AND (pumpkin OR squash OR broccoli OR peach OR pear OR *peas OR olive OR papaya OR plum OR coffee OR okra OR asparagus OR date OR *berry OR avocado OR lentil* OR sesame) OR THIRD FIELD: qualit* AND (*pollinat* OR *bee OR *bees OR hover*) AND (cacao OR persimmon OR *fruit OR apricot OR almond OR linseed OR cherry OR artichoke OR pistachio OR lupin* OR fennel OR fig* OR mustard OR quince* OR *currant OR hops OR poppy) The string reported above is 100% equal to the string suggested by the Reviewer (see below). We are sorry for the odd wording of the original string that might have generated some confusion.

		Equivalent original string (three fields): FIRST FIELD: qualit* AND SECOND FIELD: *pollinat* OR *bee OR *bees OR hover* AND THIRD FIELD: crop* OR *bean* OR palm OR tomato OR banana OR *melon OR *apple OR grape* OR citrus OR cucumber OR brassica OR *nut OR mango OR eggplant OR sunflower OR safflower OR plantain OR pepper OR pumpkin OR squash OR broccoli OR peach OR pear OR *peas OR olive OR papaya OR plum OR coffee OR okra OR asparagus OR date OR *berry OR avocado OR lentil* OR sesame OR cacao OR persimmon OR *fruit OR apricot OR almond OR linseed OR cherry OR artichoke OR pistachio OR lupin* OR fennel OR fig* OR mustard OR quince* OR *currant OR hops OR poppy
11	- Although the authors state that they included non-English literature (not clear whether for the grey literature only or overall, please clarify) the search was conducted in English, inevitably excluding potential publication in other languages. I appreciate that it may not always be possible to screen in other languages, but for a global analysis this may be necessary.	- As we do not possess the skills to conduct a complete and robust systematic literature search in multiple languages than English, we did not specifically search the literature in other languages and on other national or local databases. However, we allowed for relevant non-English studies throughout the literature search, i.e., both for the grey and the peer-reviewed literature, when articles had at least the abstract in English. With this requirement, we set a minimum standard for the quality of the literature to be included in our reviewing process. Considering the stability of the effects sizes after the inclusion of 60 additional studies (25 with the new string and 35 published in 2021, 2022 and 2023), we are confident that our results reflect well the scientific evidence on the topic.
12	- If there is a sound rationale for all the above points on the search string, it needs to be better clarified in the methods. Otherwise, I'd recommend revisiting the search string and re-run it. I accept that no search string will capture everything, but I believe that as it is, this search strategy has missed relevant articles.	- We are thankful to the Reviewer that helped us to improve the literature search. This opportunity also allowed us to incorporate the most recent studies that were initially outside the time span of the study. Please find below a Venn diagram showing the number of studies found and included with the previous and updated search string.  With the new literature search, PRISMA flowchart was updated accordingly. For full transparency, we have specified that the studies included in our analysis derive from two separate literature surveys, using different search strings. To make our search replicable we report both strings, and the study selection process of both literature

		surveys (please refer to Supplementary Fig. 6 and Supplementary Fig. 7).
13	- L247 It is not clear how the categories for the moderator “Pollinator group” were chosen. What is the definition of “natural/ambient pollinators”? Does the “natural” category include studies where pollinators were not identified to species/genus level? (this is only mentioned in the results L67 needs to be in methods). If so, I am not sure it is worth having a “other” category. Some of the species in the ‘other’ group may well overlap with natural/ambient species and they would be natural/ambient pollinators too. The only difference between “other” and “natural” seems to be that the latter were not identified, so perhaps it would make more sense to pull them together? were they not even identified by order/genus? Given that the natural community show significant effect on quality in the pollination deficit, it would be interesting to disentangle that. It may be that I am missing something and there is a sound reason for this classification, but the categories need to be defined and the rationale for choosing them be more explicit.	- Thank you for this constructive comment. We have better defined the pollinator categories and the rationale for choosing this classification (please refer to LL 296-299). “Natural pollination” included studies where there was an open pollination treatment with no info on the pollinator taxa. These studies are expected to quantify the effect of a pollinator community. To avoid confusion, we changed the name in “Pollinator community”. “Other” included only studies testing single species. As it is well-known the positive effect of having a diverse community vs. a single species, we kept the categories separated but we changed again the name to avoid confusion. ‘Other’ was changed in “Other single species”. To try to infer more info on the taxa comprising the pollinator community we reported for each crop info about the main pollinator groups (insects, mammals, birds, bats etc.) following Klein et al. 2007 (Proc. R. Soc. B. 274303–313. http://doi.org/10.1098/rspb.2006.3721). Our “pollinator community” category also includes these animals, but the exact identity of the pollinating species could not be retrieved from the studies (please refer to Supplementary Table 1). We have added this in the section on knowledge gaps (please refer to LL 168-170).
14	-Results L70-101 - Figure3 One of the pros of using the log RR is that it can be converted to a percentage, which is more intuitive to the reader. I’d suggest reporting those percentages, at least the key ones in the results. The AIC in itself is not highly informative. I’d also suggest to report the results as percentage in Figure 3, which again would aid interpretation and impact of your key results.	- We removed the AIC and we added percentages for the main results in the main text. We have changed the figures reporting effect sizes using percentages. This has greatly increased the clarity of the results.
15	-Discussion What I believe is missing in the discussion are key research gaps. For example, crops that are understudied or not studied at all? were there understudied regions? If really there is nothing on vertebrate pollinators effect on quality, this is an important gap given the economic importance of some of vertebrate pollinated fruits. These should be highlighted along recommendations on future research directions.	- Thank you again for this very useful suggestion. We have added now a research gaps and future directions section (please refer to LL 157-173)
	Minor comments	

16	- L18 Overall?	- Incorporated in the revised draft
17	- L43 (But see)?	- Incorporated in the revised draft
18	- L56 the contribution of animal pollination OR the contribution of pollinators (here and throughout the manuscript, e.g. L105-108-114)	- Incorporated in the revised draft.
19	- L60 wind/self pollination	- The pollinator exclusion treatment always allowed for wind pollination. We changed any reference to this treatment with “pollinator exclusion” to be consistent with the definition of pollination metrics presented in the Methods.
20	- L131 It would be interesting to expand and explore potential reasons why that is. Is there a threshold of pollination above which supplementary pollination does not affect quality? A saturation point? Is there any literature on this?	- Most of the experiments on pollination deficit select hand pollination as a benchmark for optimal pollination. Ideally, for many plants that benefit from outcrossing, the level of fecundity achieved by hand pollination with outcross pollen represents a perfect pollination. By manually providing pollen, saturation occurs because all the ovules are fertilised. This is not always the case in animal pollination, because some pollinators may have low pollen deposition rates (Thomson, J. D. 2001. Using pollination deficits to infer pollinator declines: Can theory guide us? Conservation Ecology 5(1): 6). We have added this text also in the Methods (please refer to LL 224-228).
21	- L134 to my understanding this is what was defined as ‘crop dependence on pollination’ and was not considered in this study. How was the 32% calculated?	- This is the result of summing up the 24% related to the service and the 8% related to deficit. We have removed this figure from the text to avoid confusion.
22	- L172 friendly land uses I a bit general, perhaps this can be expanded. Insect friendly farm management? Landscape conservation?	- We have reworded the sentence as follows (please refer to LL 195-198): “...there is the urgent need to adopt effective local and landscape management strategies to increase floral and nesting resources and to reduce current environmental pressures on wild and managed pollinators across agricultural landscapes”. We did not expand more this argument since our study does not address this topic.
23	- L179 could you add a line to briefly describe the aim of the search and some key words? Something along the line of “we conducted a systematic review....to investigate..” “we used a combination of terms..”. To give a clear indication of what the aim of the review is	- We briefly described the aim of the literature search (please refer to LL 201-202), and included key terms used in both our surveys (please refer to LL 206-208).
24	- L180 I'd state the date as from-to. The exact date of last search can be moved to the SI	- Incorporated in the revised draft. We now provide the dates of both literature searches with a from-to year format. The exact dates of both searches are specified in Appendix 1 in the Supplementary Information and in the PRISMA diagrams in Supplementary Fig. 6 and Supplementary Fig. 7.
25	- L182 Is this OR between quality crop and fruit? Clarify	The intended operator is “OR”. With this string we mean “pollinator AND quality AND (crop OR fruit). The Boolean AND command is automatically implied in all Google searches.
26	- L189 It is not clear to me what an unduplicated relevant reference is, could you clarify this?	- We are sorry for the unclear sentence. We supplemented the initial list of studies with their relevant references, that were not included in the results obtained from the Scopus, WoS and Scholar search. We also checked the references

	Did you snowball the reference from relevant papers?	of reviews, books, and meta-analyses retrieved with the databases search (but excluded based on 1 st inclusion criterion) and supplemented our list with new relevant studies. We made this clearer in the Methods (please refer to LL 247-250) and in the PRISMA diagrams (please refer to Supplementary Fig. 6 and Supplementary Fig. 7).
27	- L192 The inclusion criteria should provide a definition of animal pollinators, did you include all? (see major comment above)	- We have now followed the approach of Ratto et al. 2018 (Front Ecol Environ 2018; 16 (2): 82– 90, doi: 10.1002/fee.1763) to include a full list of major groups of pollinators including both vertebrates and invertebrates. We specified the use of the general term pollinat* to include other potential pollinating animals. We included all animals, which were tested as potential pollinators in the experimental studies assessed. We made this clearer in the main text (please see LL 237-238).
28	- L193 Animal pollination dependence OR pollinator dependence	- We removed this sentence from the text.
29	- L205 I would move this above to introduce the treatment types before the inclusion/exclusion criteria	- Incorporated in the revised draft. To provide a clearer structure to the Methods chapter, we rearranged the original paragraphs. As suggested, we moved the paragraph “definition of pollination metrics” before the inclusion criteria. We split the original “literature search and inclusion criteria” paragraph into two separate paragraphs, merging the second with the “data extraction, effect size calculation and weighting” paragraph. Thus, the first 3 new paragraphs are: i) Systematic literature search; ii) Definition of pollination metrics; iii) Literature selection process and effect size calculation.
30	- L216 spell out the acronym on first use	- Incorporated in the revised draft.
31	- L222/225 animal pollination or pollinator effect (here and throughout the manuscript)	- Thank you for the suggestion. We changed it to “animal or hand pollination”. Here, we do not necessarily mean animal pollination, as the metric can assess also the effect of hand pollination compared to animal pollination. However, we paid attention in specifying “animal pollination” throughout the manuscript, when necessary.
32	- L262 this is indeed an important moderator, but it would call it “experimental scale” or “experimental manipulation level” rather than spatial scale, which to many would evoke landscape scale vs farm scale etc	- Incorporated in the revised draft.
33	- L277 replace ‘initially’ with ‘earlier’	- Incorporated in the revised draft.

RESPONSE TO THE REVIEWER 2'S COMMENTS

	Reviewer 2's comments	Authors' response
1	- The manuscript deals with the effect of animal pollination on fruit quality using a meta-analytical approach spanning across various pollination dependent food crops globally. The authors found that pollination generally increases food quality but when broken down into specific characteristics only some traits were influenced positively. All pollinators contributed positively to quality with bumblebees being most effective. The authors also found that crop production generally suffers from a pollination deficit for wild bee communities.	
2	- Validity In general, the authors provide a valid and robust interpretation of the data. However, I have some (major and minor concerns) about the following specific interpretations.	- We are grateful for the constructive comments provided that have greatly improved the quality of the study. We have incorporated all the comments in the revised version.
3	- The title appears to be slightly misleading. It might be understood so that nutritional food is only produced by inadequate pollination. Hence, I would recommend a minor revision of the title so that this becomes clearer.	- We modified the title to avoid this potential confusion. The new title is "Inadequate pollination compromises the quality of food crops: A global meta-analysis". The new title reflects the now significant effect of pollination service also on nutritional value. Although the effect size is much smaller than that of organoleptic traits, we think that the title better reflects the main findings of this study.
4	- The conclusions of the Abstract (lines 23-26) do only fit parts of the results and the manuscript such as it deals with pollination effects and not with imperfection of fruits leading to food waste.	- We revised this part (please refer to LL 23-26). The conclusions of the Abstract are now as follows: "As producing commercially suboptimal fruits can have multiple negative economic and environmental consequences, safeguarding pollination services is an urgent priority to maintain food security".
5	- Line 45: It is not only the consumer but aspects such as shelf life (related to firmness) influences also the value for farmers and retailers.	- We removed the reference to consumers to make the sentence more general (please refer to LL 46-47)
6	- Line 49: Food quality is not only an ultimate factor for farmers income but also a first-hand factor, for instance such as farmers are paid higher for products of class 1 compared with class 2.	- Incorporated in the revised draft.
7	- Lines 90-91: This result is somehow trivial such as greenhouse growers supply sufficient managed bees to not run into pollination deficits.	- True but this is the first global study quantifying pollination deficit also under open field conditions.
8	- Lines 104-106: This sentence might have to be toned down such as there are a lot of studies that focused on crop quality but not on the scale of a meta-analysis.	- We reworded the sentence as follows (please refer to LL 111-113): "We quantified for the first time the effect of animal pollination on food quality at the global scale and across all major crops".
9	- 144-146: The discussion would benefit by adding one or two more sentences on this, also including interspecific complementarity between pollinator, such as for instance shown by Brittain et al. 2013,	- We have expanded our dataset including more studies following Reviewer 1's suggestions. With the new dataset there is no difference between pollinator groups anymore, and we removed this part from the discussion.

	Global Change Biology (https://doi.org/10.1111/gcb.12043).	
10	- 150-173: It is not just imperfection but also shorter shelf life (and other potential trait-offs), such as shown by Klatt et al. 2013, Agricultural and Food Security and Klatt et al. 2014, Proceedings B (both already cited in the reference list). It is not clear if this problem applies for all fruits but as stated in the manuscript earlier, firmness is an essential factor being determined by pollination and firmness is also a main factor for the decay of many agricultural products. Hence, this part of the discussion is completely unilateral towards consumers' behaviour but will have to include also other aspects that come with the reduced quality characteristics shown in the results. This also applies for the concluding sentences in the Abstract.	- We have now expanded the discussion including also other aspects of food quality reduction, citing both Klatt et al. 2013, Agricultural and Food Security and Klatt et al. 2014, Proceedings B, and highlighting more the direct effects on food producers, beside the effects on consumers (please refer to LL 174-193).
11	- Significance These results provide a significant inside for the field of pollination effects on food quality and thereby make an important contribution to our understanding about the importance of pollination for agricultural production.	- Thanks for the appreciation.
12	- Data and methodology The authors provide an impressive dataset derived from very many studies and covering large parts of global crop production areas. Literature research, data inclusion/exclusion are well-motivated and justified. The presentation of the data is of high quality including easy-accessible figures.	- Thanks for the appreciation. We have improved the search string following Reviewer 1's comments, and we have better clarified several aspects of the methodology.
13	- Analytical approach The statistics (from the linear mixed model point of view; please see also 'Your Expertise' section) are valid and well-motivated. I have a few questions concerning some of the analyses.	- Thanks for the appreciation.
14	- Line 72: How natural is the pollinator community of 'natural pollinated plants'? Nowadays honeybees are ubiquitous in agricultural landscapes (as it is also stated in lines 140-144). Is there any information available about the proportion of honeybees within the pollinator communities? Maybe this could be added into the models? Also, what was the proportion of honeybees in the natural communities that had decreased pollination services?	- We agree with the Reviewer and modified accordingly the moderator name replacing "natural" with "Pollinator community". The studies that sampled both honeybees and wild pollinators with a quantification of abundance were only a very small subset. Hence, it was not possible to include this variable as moderator.
15	- Lines 82-83: Looking at figure 3, the difference between bumblebees and the natural community seems hardly be significant (confidence intervals overlapping quite distinctly). It is stated that the difference was tested but no test results are shown.	- We removed this result from the text. The difference was no longer significant with the updated dataset. We report here the new results: contribution to fruit quality by bumblebees (reference) (mean=27%, CI=17% : 39%); honeybee (mean=21%, CI=12% : 30%; p=0.194); pollinator community (mean=21%, CI=14% : 28%; p=0.178)
16	- Lines 125-128: Do control for a potential bias provided by that specific crops that are pollinated by specific pollinator groups (naturally or artificially	- We provided correlations/associations between moderators in the Supplementary Figures 2 and 3 using contingency tables to show that some combinations of moderators are

	provided) have been the focus in the literature more frequently?	missing. This helps explain some of our results but does not allow us to tease apart the pure effect of some moderators. In the specific case, greenhouse studies mostly tested the effect of supplementing bumblebees.
17	- Lines 207-209: How were differences in pollination dependence between crop varieties handled? Were differences between varieties included into the calculations (lines 212-214)?	- To our knowledge, information about the pollination dependence of different varieties within a crop is missing in the literature, at least for most crops included in our analyses. We have added the importance of the variety effect in the study limitation section (please refer to LL 163-167).
18	- Lines 226-227: Does this mean that specific studies (how many?) and/or specific parameters (which?) had to be excluded?	- This can be seen in the PRISMA diagram (please refer to Supplementary Fig. 7). Since we allowed only for weighted models, studies with missing means, replicates and/or a variability measure (i.e. anything that prevented from calculating lnRR and its variance) were excluded. The number of studies that did not meet these criteria was 85 out of 218 in our last literature search. No trait was excluded at this stage, nor later in the analysis. All tested traits were at least present in one of the studies included.
19	- Lines 272-274: Such as habitats can differ largely also between latitudes, landscape type might be a good additional category, maybe describing the main landscape element in the surrounding of the study (e.g. forest, agriculture, urban, etc.).	- Unfortunately, this info was not available in most studies.
20	- Lines 276-278: Based on declining pollinator populations, could this effect not also be vice versa, i.e. larger effects in newer studies, at least for outdoor studies with natural pollinator communities?	- We have added this sentence in the Methods: "However, also the contrary is possible where a temporal increase in effect size could be related to a true decline in pollination service related to increasing environmental pressures on pollinators".
21	- Modelling in general: Where models tested for spatial and temporal autocorrelation and singularity?	- The spatial bias in meta-analysis is usually present as most studies come from a few regions. Hence, the spatial dependence is often addressed by including appropriate random factors (e.g. region ID). The temporal autocorrelation was tested by including time as moderator (please refer to LL 322-323 and Supplementary Fig. 10).
22	- Lines 330-331: A p-value <0.001 is indicating more than just 'potential bias'.	- There are several methods for identifying publication biases, and only recently new methods have been proposed for multi-level meta-analyses including random factors. After an in-depth search on the best methods to test biases with multi-level meta-analyses, we followed the recent approach proposed by Nakagawa et al. 2022 (Methods in Ecology and Evolution, 13, 4 – 21. https://doi.org/10.1111/2041-210X.13724). These authors are the most authoritative in the field, and they suggest avoiding using correlation tests as it is impossible to correct for

		effect size dependence. In fact, in the previous version our correlation between SE and sample size was very small but still significant because it used the number of effect sizes as independent observations. The modified Egger's method proposed by Nakagawa et al. 2022, on the other hand, can handle effect size dependence. The method has been recently used in other publications in high-profile journals (e.g. Bishop et al. 2022 (Ecology Letters, 25, 2034 – 2047. https://doi.org/10.1111/ele.14069); Capilla-Lasheras et al. 2022 (Ecology Letters, 25, 2552 – 2570. https://doi.org/10.1111/ele.14099)). With this method no bias emerged in our datasets. We have added this information in the methods and in the Supplementary materials. The other analyses included in the previous version were kept since they are still supported in the literature (Nakagawa et al. 2022). We thank the Reviewer for this comment since it greatly helped us to improve this important part of the study!
23	- Suggested improvements The authors argue with the commercial aspects of crop quality at various places and therefore it would add to the argumentation if monetary values could be provided as well.	- Thank you for this suggestion. We have now included this argument in the discussion. We provided a few examples of crops for which this value is well-known, and we discussed the need for more research to integrate multiple aspects of quality in the calculations (please refer to LL 183-188).
24	- Lines 116-119: Because this manuscript will be interesting for a broad readership, the discussion would benefit from some more details here so that the underlying processes can also be understood by non-expert readers on this topic.	- We have expanded the discussion about the major physiological mechanisms underlying the observed pattern. We did not go into too much details since the journal targets a very broad and general audience (please refer to LL 125-134).
25	- Further, it would be good to test for deficits between different groups of crops, for instance seed and oil crops versus fruits versus vegetables. - It would also add an interesting aspect to the results to show pollination deficits in dependence on region, country or even the intensity of agriculture. It is indicated that these results are existing in lines 122-124 (and are partly available in the Supplementary Information). This information would also be interesting in the light of the statement in lines 137-139.	- We included crop type as a moderator, following the classification adopted by Klein et al. 2007 (Proc. R. Soc. B. 274303–313. http://doi.org/10.1098/rspb.2006.3721). However, our datasets are unbalanced towards some crop types (fruits and vegetables). We report the effect of pollination service and deficit on all crop types in the main text, and we discuss the underrepresentation of some of them in the study limitations and knowledge gap paragraph (please refer to LL 159-162). - As for pollination service, we have reported the results of all models also for the pollination deficit dataset. Alongside the effect of pollination deficit on fruit quality according to region (based on latitude; results in the main text), we tested the effect of pollination deficit in relation to the contribution of agriculture to each country's GDP (Agriculture, Forestry and Fishing Value Added, share of GDP US\$, data FAO). We did not detect an effect of pollination

		deficit affecting agriculture-dependent countries (Figure 1 below).
26	- Clarity and context The manuscript concisely and clearly written. I have some questions on parts of the text that were not clear to me.	- Thanks for the appreciation.
27	- Line 33: Maybe change 'life support' to something like 'food provision' or similar.	- Incorporated in the revised draft.
28	- Line 35 Indicate what kind of deficits.	- We specified that risks of pollination deficits might affect agricultural production and human health (please refer to LL 35-36).
29	- Line 71-72: In the paragraph before you stated that effects from differences between wind and natural pollination as well as between artificial and natural pollination was assessed. Please clarify the term 'non-pollinated vs. naturally pollinated plants' within this framework.	- We reworded the sentence to avoid terminological confusion (please refer to LL 78-79).
30	- Line 109: Change 'erosion' against a less strong word, 8 % is more a decline.	- We clarified that we only detected weak signs of pollination deficits (please refer to LL 115-117).
31	- Line 113: Should it be not 'quality' itself and not 'quality improvement', because the assumption is that sufficient pollination should be available? Hence 24% of crop quality is provided by pollination or pollination contributes to 24% of crop quality.	- Thank you for advising us this clarification, we incorporated it in the revised draft.
	- Lines 130-131: Does maximised artificial pollination include greenhouse and crops where honeybees or bumblebees have been placed at the field? Was this information available for all studies under open field conditions?	- No, artificial pollination included only hand pollination. We are sorry for the inaccurate wording. We better defined in the Methods how maximized pollination was achieved in the Methods and we have changed artificial to hand pollination throughout the text (please refer to LL 223-228).
32	- Line 131: You mean exclusion of animal pollinators? Please clarify at this and other places in the manuscript.	- We replaced artificial pollination with hand pollination throughout the manuscript.
33	- Lines 134-136: This would also fit into the results section. Please also provide more information about how this value was calculated.	- We removed this sentence to avoid confusion as our meta-analysis did not include studies testing only for pollination dependence. The value was calculated by summing 24% contribution to service + 8% related to deficit. However, the studies that estimate the two values were different. Hence, we removed this value from the text.
34	- Lines 139-140: In the beginning of this paragraph and the results, it is stated that there is pollination deficit of 8% based on quality traits and 'natural' pollination but here it is stated that there is about one fourth loss. Or is this meant potentially at a complete absence of pollination?	- See above. The one-fourth was related to the 8%/32%. Now we stick to the service contribution and deficit that are clearly mathematically defined in the text (Figure 1 in the main text).
35	- References Besides my concerns stated in the different sections above, the cited literature is appropriate.	
36	Your expertise	

	Meta-analytical statistical approaches are outside my expertise. But I reviewed the statistics from the point of linear mixed modelling.	
--	--	--

Figure 1 - Effect of pollination deficit on quality of crops cultivated in agriculture-dependent countries.

Reviewer comments, further round

Reviewer #1 (Remarks to the Author):

The authors have thoroughly addressed all the concerns and suggestions that I highlighted in my original review. Changes have been made where necessary and an exhaustive response has been given to all points raised.

The manuscript has greatly improved in quality and clarity, the figures are much clearer and of high quality.

I fully agree that search strings in systematic reviews need to find the right compromise between specificity and sensitivity and there isn't such a thing as a perfect search string. However, the authors have performed a thorough exercise to find the best compromise and I am satisfied that now this meta-analysis is representative of the most relevant literature. This is a timely and needed piece of research and I would recommend it for publication.

Fabrizia Ratto

[Editor's note: Reviewer 1 also checked the authors' response to the comments from Reviewer 2, as the latter was not available in the second round]

DETAILED RESPONSE TO COMMENTS

RESPONSE TO THE REVIEWER 1'S COMMENTS

	Reviewer 1's comments	Authors' response
1	- The authors have thoroughly addressed all the concerns and suggestions that I highlighted in my original review. Changes have been made where necessary and an exhaustive response has been given to all points raised. The manuscript has greatly improved in quality and clarity, the figures are much clearer and of high quality. I fully agree that search strings in systematic reviews need to find the right compromise between specificity and sensitivity and there isn't such a thing as a perfect search string. However, the authors have performed a thorough exercise to find the best compromise and I am satisfied that now this meta-analysis is representative of the most relevant literature. This is a timely and needed piece of research and I would recommend it for publication. - Editor's note: Reviewer 1 also checked the authors' response to the comments from Reviewer 2, as the latter was not available in the second round	- We sincerely thank the Reviewer for the time and effort dedicated to reading our manuscript again, and for checking our response to the comments from Reviewer 2 as well. We thank the Reviewer for the appreciation of our work, and we are pleased that the Reviewer is happy with our revision and deem our manuscript publishable.